# Low-Carbohydrate and Low-Fat Diet with Metabolic-Dysfunction-Associated Fatty Liver Disease

**DOI:** 10.3390/nu15224763

**Published:** 2023-11-13

**Authors:** Chengxiang Hu, Rong Huang, Runhong Li, Ning Ning, Yue He, Jiaqi Zhang, Yingxin Wang, Yanan Ma, Lina Jin

**Affiliations:** 1Department of Epidemiology and Biostatistics, School of Public Health, Jilin University, Changchun 130021, China; hucx22@mails.jlu.edu.cn (C.H.); lrh22@mails.jlu.edu.cn (R.L.); heyue21@mails.jlu.edu.cn (Y.H.); zhangjiaq22@mails.jlu.edu.cn (J.Z.); yingxinwang2000@163.com (Y.W.); 2Department of Biostatistics and Epidemiology, School of Public Health, China Medical University, Shenyang 110122, China; 2022120229@cmu.edu.cn (R.H.); 2022120255@cmu.edu.cn (N.N.)

**Keywords:** low-carbohydrate diet, low-fat diet, metabolic-dysfunction-associated fatty liver disease, severe fibrosis, quality of macronutrients

## Abstract

Background: This observational cross-sectional study was designed to explore the effects of a low-carbohydrate diet (LCD) and a low-fat diet (LFD) on metabolic-dysfunction-associated fatty liver disease (MAFLD). Methods: This study involved 3961 adults. The associations between LCD/LFD scores and MAFLD were evaluated utilizing a multivariable logistic regression model. Additionally, a leave-one-out model was applied to assess the effect of isocaloric substitution of specific macronutrients. Results: Participants within the highest tertile of healthy LCD scores (0.63; 95% confidence interval [CI], 0.45–0.89) or with a healthy LFD score (0.64; 95%CI, 0.48–0.86) faced a lower MAFLD risk. Furthermore, compared with tertile 1, individuals with unhealthy LFD scores in terile 2 or tertile 3 had 49% (95%CI, 1.17–1.90) and 77% (95%CI, 1.19–2.63) higher risk levels for MAFLD, respectively. Conclusions: Healthy LCD and healthy LFD are protective against MAFLD, while unhealthy LFD can increase the risk of MAFLD. Both the quantity and quality of macronutrients might have significant influences on MAFLD.

## 1. Introduction

Nonalcoholic fatty liver disease (NAFLD) is characterized by the presence of at least 5% hepatic steatosis (HS) and the absence of other causes of liver disease [1]. This condition has become a global public health concern due to its increasing prevalence, affecting over 25% of the adult population [2]. Recently, a suggestion from a team of experts assembled from 22 countries declared that metabolic-dysfunction-associated fatty liver disease (MAFLD) was a more suitable overarching concept, reflecting pathogenesis more accurately, and avoiding the appearance of trivialization and stigmatization [3,4,5]. Compared to NAFLD, MAFLD is characterized by hepatic steatosis and metabolic dysfunction, and it also takes into account the presence of secondary causes of steatosis, such as excessive alcohol consumption [3,4,6].

MAFLD not only affects liver health, having become one of the primary causes of hepatocellular carcinoma (HCC), but also increases the risk of different kinds of diseases [1,7,8]. However, MAFLD still has no approved drug therapy. This means that it is particularly important to treat MAFLD by changing the patient’s lifestyle [3,5]. It has been proved that a changed diet is supposed to make positive sense when preventing and managing chronic diseases [9,10,11]. In recent years, restrictive dietary patterns like carbohydrate or fat restriction have been widely studied [12,13]. As it turned out, LCD and LFD were beneficial for disorders in insulin, hepatic steatosis, and cardiovascular disease (CVD) [14,15,16]. In addition to quantity, some studies reported that the quality of macronutrients exerted a significant influence on human health [17,18,19]. Some studies illustrated that the quantity and quality of diet lipids may potentially exert effects on the pathogenesis of NAFLD [20,21]. Another study demonstrated that a healthy LCD has the potential to improve hepatic steatosis by regulating metabolic levels [22].

Emerging evidence has supported a link between LCD and LFD and human health outcomes, but little information concerning the relationships between these dietary patterns and MAFLD has been known. Meanwhile, given the growing burden of MAFLD, investigation of the association between these dietary patterns and MAFLD has become imperative. Therefore, our study is designed to explore the effects of LCD and LFD scores on MAFLD. Data from the National Health and Nutrition Examination Survey (NHANES) was used for analysis.

## 2. Materials and Methods

### 2.1. Study Design and Participants

Our study made use of information about participants obtained in a 2017–2018 survey, which was the first cycle to assess liver condition using an ultrasound and vibration-controlled transient elastography (VCTE)-based FibroScan device. NHANES, a nationally representative multistage observational study, has been implemented to address the nutritional and health conditions of the non-institutional population living in the US [23]. More specific information on the NHANES can be obtained on a related website, (http://www.cdc.gov/nchs/nhanes.htm (accessed on 23 February 2023)). For the analyses, the participants aged ≥ 20 were included (*n* = 5569). After exclusion of the participants with missing dietary data (*n* = 827), incredible total energy intake (*n* = 75; (<600 or >3500 kcal/d in women, and <800 or >4200 kcal/d in men)), missing CAP data (*n* = 291), a lack of a complete elastography exam result (*n* = 294), and ineligible biological data (*n* = 121), 3961 participants were ultimately involved. See details in Appendix A.

### 2.2. Assessments of LCD and LFD Scores

Detailed diet information could be obtained using at least one valid dietary recall; usually, the first collection took place at the scene of the survey, and the second dietary recall was carried out by a later telephone interview [24]. The National Cancer Institute (NCI) method was used to estimate the usual intake of nutrients [25]. For this study, the dietary composition was based on macronutrients, including carbohydrates, protein, and fat. The major food groups have been presented in detail (see the Appendix A). Briefly, the three macronutrients were further grouped into animal and plant protein, saturated and unsaturated fat, and high- and low-quality carbohydrates based on food sources and quality [13]. We calculated the energy percentage offered by fat, protein, and carbohydrates. According to the percentage contribution to total energy, fat and protein intakes were classified into 11 equal groups in ascending order, whereas carbohydrate intake level was ranked in descending order using the same approach. The overall LCD score was determined by assigning positive scores (ranging from 0 [the lowest intake] to 10 [the highest intake]) to the total fats and total proteins, while a reverse score (ranging from 10 [the lowest intake] to 0 [the highest intake]) was assigned to the total carbohydrates. Then, adding up the scores of the three macronutrients, the final score was established, which ranged from 0–30. The higher the score, the closer people were to the overall LCD pattern [13]. Correspondingly, the “unhealthy LCD” score was determined by taking into account the consumption of high-quality carbohydrates, saturated fat, and animal protein, while low-quality carbohydrates, unsaturated fat, and plant protein were used to count up the “healthy LCD” score. Moreover, an unhealthy LCD score was characterized by a lower intake of low-quality carbohydrates, and a healthy LCD score was characterized by a lower intake of high-quality carbohydrates. Meanwhile, the LFD scores were created similarly (see Appendix A). Finally, each participant was assigned six diet scores; detailed information of the correlation matrix between diet scores can be observed in Appendix A.

### 2.3. Assessment of MAFLD

MAFLD was specifically determined by the existence of metabolic risk factors and hepatic steatosis. Generally, controlled attenuation parameter (CAP) and liver stiffness measurement (LSM) by VCTE were applied to measure liver steatosis and fibrosis, a method which has been described elsewhere [26]. A median CAP of 248 dB/m or higher would be classified as significant steatosis [27,28]. Besides hepatic steatosis, metabolic dysfunction was a vital characteristic of MAFLD; examples included overweight/obesity, the presence of type 2 diabetes(T2D), and metabolic dysregulation [3,29]. The detailed information can be found in the Appendix A.

### 2.4. Covariates Assessment

Demographic information and lifestyle-based data were collected from the questionnaire, including age, gender, race/ethnicity, education, marital status, physical leisure activity, and smoking and drinking status. Two educational levels were defined (High school or below, and Beyond high school). Marital status was divided according to Married/living with a partner or Other. Family income-to-poverty ratio (PIR) was classified into two groups (1.85<, >=1.85). Physical leisure activity assessment was based on meeting health recommendations for physical activity (at least three times of vigorous activity or five times of moderate activity during leisure time per week) [30]. Both smoking and drinking status were classified into two groups according to current behaviors. The calculation of the total energy consumption was based on the 24 h recall information about dietary intake. Hepatitis B surface antigen was used to ascertain the status of hepatitis B virus (HBV) infection, whereas detection of both hepatitis C virus (HCV) antibodies and ribonucleic acid positivity were employed to identify HCV infection [22].

### 2.5. Statistical Analysis

Given the complex sampling of NHANES, the complex weight methodology was applied in all the analyses. Weighted mean ± standard error (mean ± SE) was applied to express continuous variables, and weighted linear regression was used to make the comparison. Categorical variables were presented by the unweighted cases and weighted percentage [*n* (%)], and the comparison adopted the chi-square test. All LCD and LFD scores were divided into tertiles to compare the different effects on MAFLD, and medians were used to present the distribution of the tertiles. The Pearson’s correlation coefficient (r) was conducted to assess the association between dietary scores. The logistic regression was used to access the odds ratio (OR) and 95% confidence interval (CI) of MAFLD for LCD and LFD scores.

Model 1 was adjusted for age, gender, and race/ethnicity. Model 2 was built on Model 1, adding adjustments for educational level, marital status, PIR, physical activity, drinking status, smoking status, and total energy intake. We defined a continuous variable from a median value of each category to assess the line trend. Also, we estimated the ORs for MAFLD with a 5-point increase in each score. Additionally, the leave-one-out model was used to evaluate the association with ORs of MAFLD for substituting 3% energy from specific macronutrients for carbohydrates.

Stratification analysis was further conducted by including age, gender, race/ethnicity, current smoking, current drinking, and recommended physical activity. Meanwhile, we examined interactions between the diet scores and the subgroup variables. Considering that a great many tests would lead to a higher possibility for type I error, the Bonferroni correction was employed; the adjusted *p* value was <0.001.

The stability of the results was tested via several sensitivity analyses. Firstly, we further excluded the participants with hepatitis virus. Secondly, we presented ORs according to tertiles of macronutrient intake. Thirdly, analysis was performed to exclude participants with CVD.

All the analyses were executed with SAS statistical software (version 9.4; SAS Institute Inc., Cary, NC, USA), IBM SPSS Statistics 24.0 (IBM, Asia Analytics Shanghai) and R statistical software (version 4.2.1; R Core Team). A two-tailed *p* value < 0.05 was considered significant.

## 3. Results

### 3.1. Participant Characteristics

In total, 3961 US adults were included in this research, and the weighted prevalence of MAFLD was 56.45%. Table 1 and Table 2 describe the baseline characteristics of participants, together with associated tertiles of LCD and LFD scores. Participants with higher scores for overall LCD score or healthy LCD and healthy LFD score had a lower weighted prevalence of MAFLD. And as the unhealthy LCD and LFD scores improved, body mass index (BMI) and waist circumference (WC) increased. Participants with higher unhealthy LFD scores are younger and have a higher prevalence of MAFLD.In contrast, those with higher healthy-LCD or LFD scores are older and also showed lower BMI, lower prevalence of MAFLD, and higher PA (Table 1 and Table 2). The details as to the correlation coefficients between diet scores can be observed in Appendix A.

### 3.2. LCD and LFD Scores and MAFLD

Details of the associations between LCD and LFD scores and MAFLD are displayed in Table 3. The observed association between the overall LCD score (tertile 3: OR, 0.90, 95%CI, 0.66–1.23) (*p*-trend = 0.484), unhealthy LCD score (tertile 3: OR, 1.38, 95%CI, 0.98–1.94) (*p*-trend = 0.063), and overall LFD score (tertile 3: OR, 1.30 95%CI, 0.88–1.93), (*p*-trend = 0.161) and MAFLD was not statistically significant in the multivariable model. Compared with the lowest tertile of unhealthy LFD scores, the ORs (95%CI) of MAFLD were 1.49 (1.17–1.90) in tertile 2 and 1.77 (1.19–2.63) in tertile 3 (*p*-trend = 0.004). In addition, when compared to tertile 1, individuals in the highest tertile of diet scoreswere linked to a reduced risk of MAFLD for both healthy LCD score and healthy LFD score, with ORs of 0.63 (95%CI, 0.45–0.89) and 0.64 (95%CI, 0.48–0.86) (All *p*-trend < 0.05), respectively.

In terms of each five-point increment in diet scores, there was a positive association between unhealthy LFD score (OR, 1.27; 95%CI, 1.08–1.49) and MAFLD, while healthy LCD score and healthy LFD score were correlated with a reduced risk of MAFLD, with ORs of 0.85 (95%CI, 0.77–0.93) and 0.89 (95%CI; 0.81–0.99), respectively.

### 3.3. Isocaloric Substitution Models

In the substitution model, replacing 3% of carbohydrates with unsaturated fat was linked to a 15% (0.85; 95%CI: 0.73–0.98) reduction in the risk of MAFLD, while replacing it with unsaturated fat and plant protein was related to a 16% (0.84; 95%CI: 0.74–0.96) alleviation of the risk of MAFLD. See detail in Figure 1.

### 3.4. Subgroup and Sensitivity Analyses

In most subgroups, the associations between LCD scores, LFD scores, and MAFLD remained consistent in the stratified analyses (see Appendix A). We further found a significant interaction between healthy LCD scores and PIR on MAFLD, with the association being more pronounced in participants with a PIR ≥ 1.85 (OR: 0.77, 95% CI: 0.68, 0.87) compared to participants with a PIR < 1.85 (OR: 0.99, 95% CI: 0.90, 1.09). In sensitivity analysis, there was still a statistically significant correlation when we further excluded individuals with the hepatitis virus (see Appendix A). Analysis after excluding participants with CVD further confirmed the association between diet scores and MAFLD (see Appendix A). The association between specific macronutrients and MAFLD is shown in Appendix A.

## 4. Discussion

This cross-sectional study demonstrated that an increased risk of MAFLD was associated with unhealthy LFD scores, whereas healthy diet scores were inversely correlated with MAFLD. Additionally, substitution analysis further confirmed these findings, as a significantly lower risk of MAFLD could be observed when replacing isocaloric intake provided with unsaturated fat or unsaturated fat and plant protein for carbohydrates.

Numerous research efforts have discussed the influence of diet on NAFLD, but few have focused on MAFLD. Both the Healthy Eating Indices (HEI) and the Mediterranean Diet (MED) are beneficial in preventing MAFLD [29]. However, little information about the effects of macronutrient intake on the progression of MAFLD could be obtained clearly. Moreover, emerging evidence has demonstrated that variance in health outcomes could be attributable to the different sources of macronutrients [13,17,18,19]. Therefore, our study utilized LCD and LFD scores constructed based on the different qualities and sources of nutrients to explore how the composition of macronutrients affected MAFLD and severe fibrosis in MAFLD patients. When not considering the quality and sources of nutrients, there is no significant association between a low-carbohydrate or low-fat diet and MAFLD. Some previous findings have been in line with our results. A randomized controlled trial proved that neither carbohydrate nor fat restriction led to a significant difference in NAFLD [11]. In addition, observational studies confirmed that the overall LCD did not significantly reduce hepatocellular carcinoma risk, and when taking into account fat and protein sources, the LCD scores correlated to risk changes in health outcomes [31]. These research efforts indicated that food sources of macronutrients should be considered while evaluating the health effects of diet.

When considering the quality and sources of nutrients, a healthy low-carbohydrate and low-fat diet has a protective effect against MAFLD, while an unhealthy low-fat diet has a harmful effect on MAFLD. This finding was consistent with previous research results. Hepatic steatosis, one of the most necessary diagnostic bases of MAFLD, has been proven to be related to diet. An analysis coming from the same cycle data of NHANES claimed that those adhering to a healthy LCD exhibited a lower risk of steatosis and that unhealthy LFD was positively related to steatosis, which was partly in agreement with our results [22]. No statistically significant change was observed between dietary carbohydrate restriction and liver fat content in an intervention study [32]. The previous evidence suggested that macronutrient quantity and quality made sense in liver disease development. Another main clinical characteristic of MAFLD is metabolism, such as cases with abnormal liver function, glucose metabolic disturbance, or obesity. People with T2D may be exposed to lower mortality when adhering to vegetable and healthy LCDs [33]. In addition, our results for substitution analyses were similar to those of other studies reporting a link between high-quality carbohydrates, unsaturated fat, and a lower risk of MAFLD [22,34]. The currently drinking population in this study was relatively large, and alcohol consumption has a significant impact on the development of liver disease and metabolic syndrome [35]. Stratified analyses based on alcohol intake showed that healthy versus unhealthy dietary scores were independent of the effect of alcohol intake on MAFLD. Furthermore, compared to participants with a PIR < 1.85, participants with a PIR ≥ 1.85 experience a stronger positive impact of a healthy LCD on MAFLD, which may be attributed to their greater ability to afford a healthy dietary structure.

The observed associations between LCDs and LFDs and MAFLD among adults with MAFLD were biologically plausible. Compared with unsaturated diets, saturated-fat-enriched diets increased intrahepatic triglyceride content, contributing to hepatic lipid accumulation and poor metabolism [36]. Whole grains, which are high-quality carbohydrates, provided rich nutritional value, and their low glycemic load potential led to better weight loss and glucose control; additionally, lower LDL cholesterol, total cholesterol, and inflammation were associated with consumption of whole-grain diets [37,38]. Aside from macronutrient quality, foods supplying plant protein, high-quality carbohydrates, and unsaturated fat were related to a lower risk of MAFLD due to their bioactive components [39,40].

In addition, research has consistently demonstrated that dietary interventions hold the potential to impact the occurrence of NAFLD by positively modulating the gut microbiota [15]. Specifically, the consumption of whole grains has been shown to regulate the composition, abundance, and activity of the gut microbiota, providing substrates for the production of beneficial microbial metabolites [41]. Conversely, diets rich in high fat and saturated fatty acids have been associated with reduced richness and diversity of the intestinal microbiota, contributing to an unfavorable metabolic state [42]. Furthermore, higher bacterial abundance in individuals has been linked to a lower risk of metabolic disorders, including obesity, insulin resistance, and dyslipidemia [43]. Microbial metabolites play a pivotal role in regulating lipid and glucose metabolism and modulating inflammation and oxidative stress, as well as reducing the likelihood of liver damage [15]. Collectively, these findings provide further substantiation for our research conclusions.

There are several strengths in this study.. Our study is based on a complex sampling survey, which enhances the generalizability of our research findings. Additionally, measures for collecting data were validated, which minimized bias. However, some limitations should be considered. First, the cross-sectional design was used to explore associations, limiting the analysis of causal relationships. Second, the dietary scores were estimated according to self-reported dietary data in this study, which implicates the issue of underreporting of food consumption. Nevertheless, a validated method for the collection of dietary data has been used to reduce errors as much as possible [44]. Third, we categorized food sources of macronutrients subjectively, which might lead to misclassification and incomplete differentiation. Fourth, considering the limitations of cross-sectional studies, it is challenging to accurately determine the causal sequence between MAFLD and all extrahepatic diseases/conditions that cause metabolic dysfunction. However, to enhance the reliability of the research results, we conducted a sensitivity analysis by excluding participants with a history of heart disease, based on their medical history [45]. Fifth, previous studies have shown that several herbal medicinal products, including green tea extract, are associated with hepatotoxicity [46,47]. However, due to limitations in the available data, the underlying mechanisms of liver injury have not been fully explored. Future studies could validate the causality and further investigate how LCD and LFD influence the progression of MAFLD by impacting inflammatory factors, oxidative stress, gut microbiota, etc.

## 5. Conclusions

In summary, healthy LCD and healthy LFD are protective against MAFLD, while unhealthy LFD can increase the risk of MAFLD. All findings support the proposition that MAFLD prevention could potentially benefit from paying attention to the quantity and quality of dietary intake. In addition, further cohort or intervention research needs to be implemented to confirm our findings.

## Figures and Tables

**Figure 1 nutrients-15-04763-f001:**
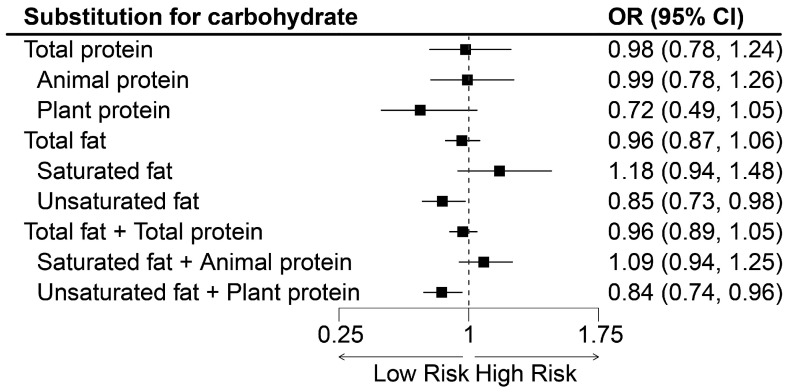
The association between replacing 3% of energy from carbohydrates with specific macronutrients and MAFLD in NHANES (2017–2018). MAFLD, metabolic-dysfunction-associated fatty liver disease. Adjusted for age (continuous), gender (male/female), race/ethnicity (non-Hispanic White/other), educational level (high school or below/college or above), marital status (married or living with partner/other), PIR (<1.85/≥1.85), current smoker (yes/no), current drinker (yes/no), recommended physical activity (yes/no), and total energy intake (continuous).

**Table 1 nutrients-15-04763-t001:** Characteristics of study participants by tertile of LCD score in NHANES (2017–2018).

Characteristic	Overall LCD Score		*p*	Unhealthy LCD Score		*p*	Healthy LCD Score		*p*
	Tertile 1	Tertile 3		Tertile 1	Tertile 3		Tertile 1	Tertile 3	
Participants, *n*	1429	1257		1423	1270		1476	1277	
Median score (IQR)	7 (4–9)	24 (21–27)		8 (5–11)	22 (20–24)		9 (6–11)	22 (20–24)	
Age, years	48.01 ± 0.88	47.63 ± 0.95	0.723	52.18 ± 0.84	44.3 ± 0.98	<0.001	44.45 ± 0.77	50.31 ± 1.23	0.001
BMI, kg/m^2^	29.49 ± 0.29	29.78 ± 0.38	0.297	28.71 ± 0.30	30.2 ± 0.35	0.012	29.58 ± 0.28	28.99 ± 0.34	0.017
WC, cm	99.72 ± 0.70	101.22 ± 0.97	0.085	98.11 ± 0.71	102.23 ± 0.94	0.006	100.01 ± 0.66	99.23 ± 0.95	0.051
Female, %	823 (57.36)	534 (43.58)	<0.001	868 (58.88)	490 (39.86)	<0.001	746 (52.22)	643 (51.05)	0.834
Race/Ethnicity			<0.001			<0.001			<0.001
Non-Hispanic White	414 (53.71)	552 (69.87)		357 (53.91)	579 (68.96)		467 (54.14)	504 (68.21)	
Other	1015 (46.29)	705 (30.13)		1066 (46.09)	691 (31.04)		1009 (45.86)	773 (31.79)	
Educational level			0.009			0.148			<0.001
High school or below	627 (41.79)	487 (32.81)		567 (35.48)	538 (38.22)		689 (46.58)	441 (30.17)	
Above high school	799 (58.21)	769 (67.19)		853 (64.52)	731 (61.78)		783 (53.42)	834 (69.83)	
Marital status			0.254			0.751			0.019
Married or living with partner	852 (59.11)	751 (63.79)		894 (62.87)	725 (60.74)		820 (56.84)	826 (64.59)	
Other	575 (40.89)	505 (36.21)		528 (37.13)	545 (39.26)		653 (43.16)	451 (35.41)	
PIR			<0.001			0.264			<0.001
<1.85	587 (37.33)	417 (24.95)		512 (31.47)	474 (28.49)		646 (40.2)	395 (23.69)	
≥1.85	666 (62.67)	703 (75.05)		734 (68.53)	673 (71.51)		650 (59.8)	750 (76.31)	
Current smoker, %	240 (19.53)	234 (15.26)	0.203	164 (13.49)	309 (19.46)	<0.001	347 (23.51)	164 (11.59)	<0.001
Current drinker, %	903 (72.24)	910 (81.33)	<0.001	871 (69.95)	949 (83.34)	<0.001	992 (73.83)	892 (77.73)	0.125
Recommended physical activity, %	307 (24.90)	345 (34.18)	<0.001	352 (28.38)	313 (28.69)	0.821	284 (20.42)	396 (38.57)	<0.001
MAFLD, %	783 (55.57)	727 (52.43)	0.613	787 (52.25)	729 (54.54)	0.672	811 (56.85)	714 (49.18)	0.032
CAP, dB/m	263.79 ± 2.40	262.67 ± 3.13	0.899	258.79 ± 2.82	264.88 ± 3.33	0.265	263.88 ± 2.02	258.77 ± 3.48	0.054
Dietary intake									
Total energy, kcal/d	2089 ± 24	1901 ± 19	<0.001	1980 ± 22	2009 ± 18	0.694	2126 ± 18	1870 ± 17	<0.001
Total carbohydrate, % of total energy intake	54.88 ± 0.10	44.03 ± 0.16	<0.001	53.51 ± 0.14	45.34 ± 0.20	<0.001	53.22 ± 0.18	45.31 ± 0.24	<0.001
High-quality carbohydrate	9.88 ± 0.32	7.97 ± 0.15	<0.001	12.8 ± 0.27	5.84 ± 0.15	<0.001	6.91 ± 0.25	10.44 ± 0.26	<0.001
Low-quality carbo hydrate	45.00 ± 0.32	36.06 ± 0.23	<0.001	40.71 ± 0.32	39.50 ± 0.32	<0.001	46.31 ± 0.28	34.86 ± 0.19	<0.001
Total fat, % of total energy intake	30.32 ± 0.12	37.75 ± 0.10	<0.001	31.22 ± 0.20	36.99 ± 0.13	<0.001	31.53 ± 0.14	36.98 ± 0.18	<0.001
Unsaturated fat	19.47 ± 0.10	24.20 ± 0.09	<0.001	20.62 ± 0.15	23.28 ± 0.10	<0.001	19.84 ± 0.08	24.16 ± 0.12	<0.001
Saturated fat	10.85 ± 0.05	13.55 ± 0.10	<0.001	10.6 ± 0.07	13.71 ± 0.09	<0.001	11.69 ± 0.07	12.82 ± 0.10	<0.001
Total protein, % of total energy intake	14.8 ± 0.08	18.22 ± 0.12	<0.001	15.27 ± 0.10	17.68 ± 0.12	<0.001	15.25 ± 0.09	17.71 ± 0.12	<0.001
Plant protein	5.25 ± 0.05	5.42 ± 0.04	0.044	5.75 ± 0.06	5.10 ± 0.03	<0.001	4.82 ± 0.03	5.91 ± 0.04	<0.001
Animal protein	9.55 ± 0.06	12.8 ± 0.13	<0.001	9.52 ± 0.07	12.58 ± 0.12	<0.001	10.44 ± 0.08	11.81 ± 0.13	<0.001

Variables are shown as the weighted mean ± standard errors or unweighted cases and weighted percentage [*n* (%)]. Abbreviations: LCD, low-carbohydrate diet; BMI, body mass index; WC, waist circumference; PIR, family income-to-poverty ratio; MAFLD: metabolic-dysfunction-associated fatty liver disease; CAP, median controlled attenuation parameter.

**Table 2 nutrients-15-04763-t002:** Characteristics of study participants by tertile of LFD score in NHANES (2017–2018).

Characteristic	Overall LFD Score		*p*	Unhealthy LFD Score		*p*	Healthy LFD Score		*p*
	Tertile 1	Tertile 3		Tertile 1	Tertile 3		Tertile 1	Tertile 3	
Participants, *n*	1449	1206		1569	1069		1380	1311	
Median score (IQR)	9 (7–11)	21 (20–23)		10 (8–12)	21 (20–23)		8 (5–10)	23 (21–26)	
Age, years	48.84 ± 0.91	46.87 ± 0.73	0.136	52.15 ± 1.02	42.1 ± 0.78	<0.001	45.17 ± 0.67	50.96 ± 0.69	<0.001
BMI, kg/m^2^	29.74 ± 0.35	29.20 ± 0.27	0.484	29.34 ± 0.35	29.67 ± 0.35	0.291	30.43 ± 0.36	28.66 ± 0.28	0.003
WC, cm	101.24 ± 0.85	98.52 ± 0.79	0.064	100.33 ± 0.98	99.8 ± 0.92	0.841	102.69 ± 0.82	97.36 ± 0.75	<0.001
Female, %	704 (47.78)	635 (55.38)	0.043	871 (54.73)	479 (48.91)	0.029	619 (46.26)	753 (59.26)	<0.001
Race/Ethnicity			<0.001			<0.001			<0.001
Non-Hispanic White	689 (74.59)	229 (40.84)		669 (72.84)	296 (48.49)		639 (69.82)	271 (45.81)	
Other	760 (25.41)	977 (59.16)		900 (27.16)	773 (51.51)		741 (30.18)	1040 (54.19)	
Educational level			0.014			<0.001			0.107
High school or below	551 (34.6)	577 (44.66)		541 (30.53)	528 (48.30)		591 (40.16)	527 (33.72)	
Above high school	896 (65.4)	624 (55.34)		1026 (69.47)	538 (51.70)		787 (59.84)	780 (66.28)	
Marital status			0.513			0.089			0.104
Married or living with partner	825 (63.28)	747 (60.61)		966 (64.81)	613 (57.53)		765 (58.66)	857 (65.13)	
Other	623 (36.72)	456 (39.39)		602 (35.19)	454 (42.47)		614 (41.34)	453 (34.87)	
PIR			<0.001			<0.001			0.062
<1.85	510 (26.10)	506 (40.62)		490 (23.42)	487 (43.44)		557 (31.82)	473 (32.52)	
≥1.85	790 (73.90)	530 (59.38)		905 (76.58)	438 (56.56)		691 (68.18)	682 (67.48)	
Current smoker, %	277 (17.18)	195 (18.83)	0.209	208 (13.38)	244 (23.25)	<0.001	364 (23.80)	129 (11.08)	<0.001
Current drinker, %	1019 (78.25)	754 (72.42)	0.035	1076 (77.61)	725 (76.56)	0.658	984 (78.72)	818 (71.20)	0.007
Recommended physical activity, %	354 (29.46)	272 (25.85)	0.363	422 (31.85)	222 (21.30)	0.002	285 (22.97)	360 (32.69)	0.003
MAFLD, %	815 (51.37)	692 (56.69)	0.250	892 (50.00)	599 (57.08)	0.013	782 (55.38)	719 (49.77)	0.033
CAP, dB/m	261.69 ± 3.34	266.54 ± 2.46	0.549	260.39 ± 3.44	266.97 ± 3.04	0.317	265.29 ± 2.42	257.29 ± 3.12	0.185
Dietary intake									
Total energy, kcal/d	2013 ± 20	1955 ± 16	0.063	1951 ± 17	2048 ± 23	0.002	2094 ± 17	1881 ± 22	<0.001
Total carbohydrate, % of total energy intake	45.38 ± 0.19	54.46 ± 0.18	<0.001	46.5 ± 0.24	53.22 ± 0.20	<0.001	47.49 ± 0.23	52.17 ± 0.20	<0.001
High-quality carbo hydrate	7.39 ± 0.21	11.00 ± 0.24	<0.001	9.93 ± 0.26	7.15 ± 0.21	<0.001	5.30 ± 0.15	13.89 ± 0.23	<0.001
Low-quality carbo hydrate	37.98 ± 0.29	43.47 ± 0.35	<0.001	36.57 ± 0.18	46.07 ± 0.36	<0.001	42.19 ± 0.34	38.29 ± 0.26	<0.001
Total fat, % of total energy intake	37.92 ± 0.09	29.14 ± 0.10	<0.001	36.95 ± 0.15	30.43 ± 0.13	<0.001	36.19 ± 0.13	31.26 ± 0.20	<0.001
Unsaturated fat	24.30 ± 0.11	18.81 ± 0.08	<0.001	24.08 ± 0.11	19.11 ± 0.08	<0.001	22.71 ± 0.1	20.74 ± 0.15	<0.001
Saturated fat	13.62 ± 0.10	10.33 ± 0.05	<0.001	12.87 ± 0.10	11.31 ± 0.07	<0.001	13.48 ± 0.07	10.53 ± 0.06	<0.001
Total protein, % of total energy intake	16.71 ± 0.12	16.40 ± 0.11	0.046	16.55 ± 0.14	16.35 ± 0.13	0.338	16.31 ± 0.13	16.56 ± 0.08	0.149
Plant protein	5.23 ± 0.05	5.53 ± 0.05	0.002	5.54 ± 0.05	5.06 ± 0.04	<0.001	4.73 ± 0.02	6.17 ± 0.03	<0.001
Animal protein	11.47 ± 0.13	10.86 ± 0.1	0.005	11.01 ± 0.13	11.29 ± 0.1	0.170	11.58 ± 0.11	10.39 ± 0.08	<0.001

Variables are shown as the weighted mean ± standard errors or unweighted cases and weighted percentage [*n* (%)]. Abbreviations: LFD, low-fat diet; BMI, body mass index; WC, waist circumference; PIR, family income-to-poverty ratio; MAFLD, metabolic-dysfunction-associated fatty liver disease; CAP, median controlled attenuation parameter.

**Table 3 nutrients-15-04763-t003:** Association of LCD scores and LFD scores with MAFLD in NHANES (2017–2018).

	Tertiles of Diet Scores			*p* for Trend	Per Five-Point Increase
	Tertile1	Tertile2	Tertile3		
Overall LCD score	7 (4–9)	16 (13–17)	24 (21–27)		
Median score (IQR)					
Cases/participants, *n*/*n*	783/1429	726/1275	727/1257		
Model 1	Reference	0.93 (0.68, 1.28)	0.87 (0.66, 1.15)	0.302	0.95 (0.88, 1.02)
Model 2	Reference	0.91 (0.67, 1.25)	0.90 (0.66, 1.23)	0.484	0.95 (0.87, 1.04)
Unhealthy LCD score					
Median score (IQR)	8 (5–11)	16 (14–17)	22 (20–24)		
Cases/participants, *n*/*n*	787/1423	720/1268	729/1270		
Model 1	Reference	1.27 (1.05, 1.54)	1.34 (1.02, 1.75)	0.035	1.09 (1.00, 1.18)
Model 2	Reference	1.21 (0.93, 1.57)	1.38 (0.98, 1.94)	0.063	1.11 (0.99, 1.24)
Healthy LCD score					
Median score (IQR)	9 (6–11)	15 (14–17)	22 (20–24)		
Cases/participants, *n/n*	811/1476	711/1208	714/1277		
Model 1	Reference	0.85 (0.66, 1.09)	0.63 (0.48, 0.84)	0.004	0.85 (0.79, 0.92)
Model 2	Reference	0.80 (0.59, 1.07)	0.63 (0.45, 0.89)	0.013	0.85 (0.77, 0.93)
Overall LFD score					
Median score (IQR)	9 (7–11)	15 (14–17)	21 (20–23)		
Cases/participants, *n*/*n*	815/1449	729/1306	692/1206		
Model 1	Reference	1.19 (0.92, 1.55)	1.25 (0.93, 1.68)	0.117	1.07 (0.97, 1.19)
Model 2	Reference	1.24 (0.91, 1.69)	1.30 (0.88, 1.93)	0.161	1.10 (0.96, 1.25)
Unhealthy LFD score					
Median score (IQR)	10 (8–12)	16 (15–17)	21 (20–23)		
Cases/participants, *n*/*n*	892/1569	745/1323	599/1069		
Model 1	Reference	1.45 (1.20, 1.75)	1.66 (1.23, 2.24)	0.001	1.24 (1.10, 1.41)
Model 2	Reference	1.49 (1.17, 1.90)	1.77 (1.19, 2.63)	0.004	1.27 (1.08, 1.49)
Healthy LFD score					
Median score (IQR)	8 (5–10)	15 (13–17)	23 (21–26)		
Cases/participants, *n*/*n*	782/1380	735/1270	719/1311		
Model 1	Reference	0.90 (0.74, 1.08)	0.64 (0.51, 0.80)	0.001	0.89 (0.82, 0.95)
Model 2	Reference	0.89 (0.71, 1.12)	0.64 (0.48, 0.86)	0.008	0.89 (0.81, 0.99)

Abbreviations: LCD, low-carbohydrate diet; LFD, low-fat diet; MAFLD, metabolic-dysfunction-associated fatty liver disease. Model 1: Adjusted for age (continuous), gender (male/female), and race/ethnicity (non-Hispanic White/other). Model 2: Adjusted for age (continuous), gender (male/female), race/ethnicity (non-Hispanic White/other), educational level (high school or below/college or above), marital status (married or living with partner/other), PIR (<1.85/≥1.85), current smoker (yes/no), current drinker (yes/no), recommended physical activity (yes/no), and total energy intake (continuous).

## Data Availability

Data described in the manuscript, in addition to the code book and analytic code will be made publicly and freely available without restriction at https://www.cdc.gov/nchs/nhanes/about_nhanes.htm URL (accessed on 23 February 2023).

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
