# Peer review of "Low-Carbohydrate and Low-Fat Diet with Metabolic-Dysfunction-Associated Fatty Liver Disease"

_nutrients, 2023, doi:10.3390/nu15224763_

Round 1

Reviewer 1 Report (New Reviewer)

Comments and Suggestions for Authors

The authors present findings that underscore the pivotal role of dietary patterns in the context of Metabolic Dysfunction Associated Fatty Liver Disease (MAFLD) and offer a novel perspective on its prevention and management. The manuscript and study are well conducted; however, there are areas that could be improved. Please find my suggestions below.

Introduction

The introduction is generally clear and well-organized. However, consider breaking down some longer sentences for improved readability.

It would be helpful to briefly provide detail on MAFLD, and how it differs form NAFLD upon its first mention for readers who may not be familiar with the term as it is a fairly recent nomenclature

The introduction sets the stage for the importance of studying the association between LCDs, LFDs, and MAFLD. Consider adding a sentence that explicitly states the research objective or hypothesis to guide the reader.

Methods:

The explanation of how MAFLD was assessed using controlled attenuation parameter (CAP) and liver stiffness measurement (LSM) is clear and concise. It's good to provide references to previous studies for further details.  

Clearly stating the criteria for participant exclusion (e.g., incredible total energy intake, incomplete elastography exam, ineligible biological data) is important for transparency.

Clearly stating what variables were adjusted for in each model (Model 1 and Model 2)

Results

The results are well presented but is bit heavy on tables and figures, based on the importance, the author may consider moving some as supplementary tables to improve readability of the paper.

Discussion

Well drafted, only suggestion would be you could add specific areas for future research to build upon the current findings possibily to establish casuality etc,

Advised to have a conclusion section. 

Author Response

Reviewer 2 Report (Previous Reviewer 3)

Comments and Suggestions for Authors

The manuscript is well-structured and presents an interesting clinically important message to manage with metabolic dysfunction-associated liver disease.

Points, still to be addressed for clearer publication;

1- inclusion and exclusion criteria of the participant should be summarized in the text under material and method, either in brief statement or  the figure in supplemental data be included in the manuscript.

2. In the abstract: It should start with: This observational cross-sectional study/research was designed to explore the effects of low carbohydrate......etc.

This is more precisely drafted and will not touch the strength of the data and conclusion.

Round 2

Reviewer 1 Report (New Reviewer)

Comments and Suggestions for Authors

The manuscript has undergone a thorough revision based on received feedback. However, the abstract and conclusion sections still lack an adequate representation of the study's extent. I would suggest a minor revision to include key points in both the abstract and conclusion, ensuring they accurately reflect the scope of the study.

Comments on the Quality of English Language

Minor errors, proof reading required.

This manuscript is a resubmission of an earlier submission. The following is a list of the peer review reports and author responses from that submission.

Round 1

Reviewer 1 Report

Comments and Suggestions for Authors

The manuscript is interesting, but the way it is presented to the reader loses interest. It also has methodological and results errors.

1) The authors should indicate whether other causes of hepatitis (autoimmune, pharmacological, etc.) apart from hepatitis B, C and alcohol intake were ruled out. It is known that overweight and obesity increases the risk of liver damage from other etiologies.

2) They do not define the median score

3) The results are not presented clearly. 4) Tables 1 and 2 please include p values; Furthermore, include  standard deviation instead  standard error. 5) In tables 1 and 2, the number of participants does not coincide with the total sum of participants from T1 and T3 of the healthy and unhealthy carbohydrates or fat diets. 6) If the ICs are analyzed, the majority are not significant., so is difficult to conclud the effect of the diet .

7) In the discussion they authors should add a paragraph about the effect of alcohol

intake in  MAFLD, because according to the results presented, 78% of the participants

drank alcohol.

8) Other causes of liver damage are not discussed, such us  the consumption of drugs or other xenobiotics (green tea Toxicol Re2020 Feb 15;7:386-402.  doi: 10.1016/j.toxrep.2020.02.008. eCollection 2020, Aliment Pharmacol Ther 2013 Jan;37(1):3-17.  doi: 10.1111/apt.12109. Epub 2012 Nov 5.; Biomed Pharmacother. 2017 Dec;96:798-811. doi: 10.1016/j.biopha.2017.10.055. Epub 2017 Nov 6.) that could be associated.9) I suggest adding a paragraph to the discussion that comments the effect of the type of carbohydrate and fat on the intestinal microbiota and how it influences the development of MALFD.

In summary, I recommend improving the presentation of the methodology, results and discussion so that the manuscript is publishable.

Reviewer 2 Report

Comments and Suggestions for Authors

The manuscript of Hu et al. is aimed at exploring the implications of low-carbohydrate and low-fat diet in the metabolic disturbances linked to fatty liver disease and fibrosis. Authors used data from publicly available database, the National Health and Nutrition Examination Survey (NHANES), in specific from the 2017-2018 period. Authors validate their time period choice as this data collection was the first one to include ultrasound and vibration-controlled transient elastography to access liver condition. In their final analysis, Authors included 3961 adults and analyzed the relationship between nutritional quantitative and qualitive parameters and the presence metabolic dysfunction associated liver disease (MAFLD) and fibrosis in a cross-sectional study. The nutritional parameters were determined by diet scores assigned to carbohydrates and fats based upon their healthy or unhealthy features. Authors performed several analyses of their data to determine odds ratios using 2 different Models with different adjustment criteria. Based upon their data, Authors conclude that an increased risk of MAFLD is associated with unhealthy diet scores, while healthy diet scores were associated with a decreased risk of MAFLD. The association between the risk of fibrosis and diet scores in MAFLD patients was less straightforward suggesting perhaps a collider bias in the analysis of the MAFLD subpopulation of study subjects. This possibility is also suggested by the very low percentage of individuals with severe liver fibrosis. Authors also address the limitations of their study. 

Authors complements the manuscript with 3 Figures, 4 Tables and Supplementary Figures, and cite 44 publications to put their findings in context. 

The manuscript fits the scope of the “Nutrients” and is of interest for the readers of the journal. The text of the manuscript is succinctly written and easy to follow. 

However, this reviewer notes the following issues that need to be addressed before the manuscript could be considered for publication.

Major issues:

1.     The title is too long and uses the “association” repeatedly, please rephrase

2.     Supplementary Figure S1: please provide more details about the values used for judging “implausible energy intake” that was used as exclusion criteria.

Minor issues:

1.     Please spell out abbreviations at their first mentioning, e.g LCD, LFD, MAFLD etc.

2.     Page 3, Line 93 “somewhere else” please change to “elsewhere”. 

3.     Page 3, Line 144: “involved” please change for “included”

4.     Page 4, Line 149: “more possibility“ please change for “higher probability”

5.     Table 1 and 2: please mention PIR among the abbreviations

6.     Page 12, Line 267: “had discussed” and “had focused” please change for “have discussed” and “focused”

7.     Page 12, Line 271: “had demonstrated” please change to “have demonstrated”

Comments on the Quality of English Language

The suggested changes are marked in the Minor points in the review. 

Author Response

请参阅附件

Reviewer 3 Report

Comments and Suggestions for Authors

In Materials and Methods: line 68, since the supplementary figure 1 displays the inclusion and exclusion criteria, it should be clearly written that inclusion and exclusion criteria for this study are displayed or presented in the supplementary figure 1.

 Most of the limitations of this study that a reviewer would think about are addressed in the discussion of the current manuscript, which make this paper more informative for future studies.

However, an additional limitation should be added in the discussion. The authors should consider citing this review: Eda Kaya and Yusuf Yilmaz (J. Clini. Transl. Hepatology 2022 Apr 28; 10 (2): 329-338. Metabolic-associated Fatty Liver Disease (MAFLD): A Multi-systemic Disease Beyond the Liver. In discussion, drafted as an additional limitation that all diseases/conditions that are known to be extrahepatic and cause metabolic dysfunction should be excluded from the study, or unless a multidisciplinary /multi-systemic screening and assessment of MAFLD is used.

Any other extrahepatic factor pathological or pathophysiological feature/condition that diagnostically or clinically cause metabolic associated fatty liver should be considered in setting the inclusion and exclusion criteria.

Pasting from the conclusion of that publication: “MAFLD must be evaluated as a multi-systemic disease affecting many extrahepatic organs. The disease burden extends beyond liver-related complications, underlining the importance of multidisciplinary screening and disease management”.

Minor: line 337: put the missing “of”: from paying attention to quantity and quality of dietary intake

Comments on the Quality of English Language

The manuscript is written in perfect English. However, polishing the text and  minor edits are recommended.

Author Response

请参阅附件
